# Conformation of Flexible and Semiflexible Chains Confined in Nanoposts Array of Various Geometries

**DOI:** 10.3390/polym12051064

**Published:** 2020-05-06

**Authors:** Zuzana Benková, Lucia Rišpanová, Peter Cifra

**Affiliations:** Polymer Institute, Slovak Academy of Sciences, Dúbravská cesta 9, 845 41 Bratislava, Slovakia; lucia.rispanova@savba.sk (L.R.); peter.cifra@savba.sk (P.C.)

**Keywords:** nanopost array, geometry variation, semiflexible chain, flexible chain, molecular dynamics, occupation number, axial chain extension, structure factor, DNA, microfluidic devices

## Abstract

The conformation and distribution of a flexible and semiflexible chain confined in an array of nanoposts arranged in parallel way in a square-lattice projection of their cross-section was investigated using coarse-grained molecular dynamics simulations. The geometry of the nanopost array was varied at the constant post diameter *d*_p_ and the ensuing modifications of the chain conformation were compared with the structural behavior of the chain in the series of nanopost arrays with the constant post separation *S*_p_ as well as with the constant distance between two adjacent post walls (passage width) *w*_p_. The free energy arguments based on an approximation of the array of nanopost to a composite of quasi-channels of diameter *d*_c_ and quasi-slits of height *w*_p_ provide semiqualitative explanations for the observed structural behavior of both chains. At constant post separation and passage width, the occupation number displays a monotonic decrease with the increasing geometry ratio *d*_c_/*w*_p_ or volume fraction of posts, while a maximum is observed at constant post diameter. The latter finding is attributed to a relaxed conformation of the chains at small *d*_c_/*w*_p_ ratio, which results from a combination of wide interstitial volumes and wide passage apertures. This maximum is approximately positioned at the same *d*_c_/*w*_p_ value for both flexible and semiflexible chains. The chain expansion from a single interstitial volume into more interstitial volumes also starts at the same value of *d*_c_/*w*_p_ ratio for both chains. The dependence of the axial chain extension on the *d*_c_/*w*_p_ ratio turns out to be controlled by the diameter of the interstitial space and by the number of monomers in the individual interstitial volumes. If these two factors act in the same way on the axial extension of chain fragments in interstitial volumes the monotonic increase of the axial chain extension with the *d*_c_/*w*_p_ in the nanopost arrays is observed. At constant *w*_p_, however, these two factors act in opposite way and the axial chain extension plotted against the *d*_c_/*w*_p_ ratio exhibits a maximum. In the case of constant post diameter, the characteristic hump in the single chain structure factor whose position correlates with the post separation is found only in the structure factor of the flexible chain confined in the nanopost array of certain value of *S*_p_. The structure factor of the flexible chain contains more information on the monomer organization and mutual correlations than the structure factor of the semiflexible chain. The stiffer chain confined in the nanopost array is composed of low number of statistical segments important for the presence of respective hierarchical regimes in the structure factor.

## 1. Introduction

Geometrically confined macromolecules are frequently encountered in a large number of biochemical processes and nanotechnological devices. The conformation adopted by a confined biomacromolecule is dictated by the strength and geometry of confinement and its knowledge, in turn, is crucial for understanding processes and phenomena occurring in living systems such as stabilization of confined proteins against reversible unfolding [1,2], reaction kinetics of biomacromolecules [3] or dynamics of DNA unknotting [4]. Thus, the control of polymer conformation induced by confinement enables manipulation of the function of biomacromolecules. The microfluidic devices fabricated for DNA mapping or for the study of static and dynamic properties of a confined biopolymer [5,6,7,8,9] are based on manipulation of the conformation of a single chain confined in a nano- or microchannel. Recent advances in microfabrication allow the construction of more complex geometrical confinements, for example, an array of collinearly organized nanoposts used in electrophoretic separations of DNA [10,11]. This array appears to be as efficient as the commonly adopted gel matrices in electrophoretic devices and its applicability for given molecular lengths is readily tunable through adjusting the arrangement, size and separation of the nanoposts as well as the applied electric field and the pulse time. Presently, the structure of flexible and semiflexible chains confined in objects of simple geometry such as a slit [12,13,14,15,16,17], channel [7,8,18,19,20,21,22,23,24,25,26,27,28,29,30,31,32,33,34,35], or sphere [36,37,38,39,40] are well explored at the theoretical, simulation and experimental level. A great deal of attention has been also paid to the translocation of a chain through a pore between interconnected cavities [41,42,43,44,45,46]. This issue is related to the translocation of biomacromolecules across biological membranes, transport mechanism of drug delivery or entropy-driven segregation of polymer chains under geometrical confinement relevant to bacterial chromosome replication and separation upon cell division. In living systems, the confinement of a biomacromolecule is often combined with its translocation. 

In this work, the structural behavior of a single chain confined in the array of infinitely long collinear nanoposts arranged in a square lattice has been studied using coarse-grained molecular dynamics (MD) simulations. Contrary to the electrophoretic setups, there are no constraining walls acting on a confined chain in the direction perpendicular to the post axes. The conformation of a polymer chain confined in this array is only partly restricted by the presence of nanoposts in the lateral direction, since the chain might translocate through the space between two adjacent nanoposts. As there are two geometric parameters, the post diameter and the post separation characterizing the geometry of a nanopost array, the geometry variation of the nanopost array might be realized in several ways. The effects arising from the geometry variation of post array at fixed post separation and at fixed distance between the walls of adjacent posts have been already investigated [47,48,49,50,51]. In this study, the geometry of a nanopost array is varied at constant post diameter and its effect on the chain conformation is compared with the effect ensuing from the geometry variation at constant post separation or constant distance between the walls of two adjacent posts. The knowledge of structural behavior of a polymer confined in a nanopost array might be useful not only for nanotechnology but also for better understanding of the structural behavior of biomacromolecules in the crowded environment of cells. In addition to the modification of the confinement strength through the geometrical parameters of the nanopost array, the stiffness of the chain backbone is the other factor which also considerably influences the conformation of a biomacromolecule confined in this array. In order to address the effect of the chain stiffness the structural behavior of both, the confined semiflexible chain, with its stiffness corresponding to a DNA at high ionic strength, and the flexible chain are simulated. The occupation number, chain span along the post axes, and single chain structure factor evaluated as functions of the geometric parameters provide the information on the structural behavior of a confined chain. It is of interest to scrutinize how the different ways of geometry modification affect the conformation of a confined chain and whether the relative or rather the absolute values of the geometrical parameters of a nanopost array govern the chain conformation and distribution in the space restricted by the posts. The effort is also developed to provide insight into the effect of the geometry of the nanopost array and the structural parameters of the confined chain on the resulting conformation of the confined chain. 

## 2. Method and Model

The present study deals with the effect of geometrical confinement imposed by a square array of parallel nanoposts on the conformation of a flexible and semiflexible polymer chains using coarse-grained molecular dynamics (MD) simulations. The discretized bead-spring worm-like chain (WLC) model has been used for the representation of a semiflexible chain. In this model, a chain consists of effective monomers (beads) consecutively connected by effective fluctuating bonds (springs). Each bead contains several atoms and thus, the atomistic details are neglected. The same model with the vanishing energy penalty resulting from the deformation of the effective angles has been employed also for the representation of a flexible chain. Since the effective monomers possess neutral charge, the component of electrostatic interactions is zero. The non-bonded interactions are composed of the van der Waals monomer-monomer and monomer-post pair interactions while the bonded interactions consider the bond stretching and the bending of two consecutive effective bonds.

The non-bonded interactions were represented by the entirely repulsive Weeks-Chandler-Anderson (WCA) potential:(1)$$U_{\mathrm{WCA}}\left(r_{ij}\right)=4\varepsilon \left[{\left(\frac{\sigma }{r_{ij}}\right)}^{12}-{\left(\frac{\sigma }{r_{ij}}\right)}^{6}\right]+\varepsilon \;$$
where the Lennard-Jones parameters *ε* = 1 and *σ* = 1 denote the interaction energy between the beads and the bead diameter, respectively, and *r_ij_* represents the distance between bead *i* and *j*. This potential was applied also to the interaction between two bonded monomers. The Lennard-Jones parameters are considered as the units of energy and length, respectively, in this study. The attractive part of WCA potential is removed by imposing the conditions $U_{\mathrm{WCA}}\left(r_{ij}\right)=0$ for $r_{ij}\geq 2^{1/6}\sigma \;$. The same kind of repulsive potential was used for the representation of interactions between effective monomers and posts:(2)$$U_{\mathrm{WCA}}\left(r\right)=4\varepsilon \left[{\left(\frac{\sigma }{r-D_{p}/2}\right)}^{12}-{\left(\frac{\sigma }{r-D_{p}/2}\right)}^{6}\right]+\varepsilon$$
where *r* is the distance between an effective monomer and a post axis and *D*_p_ is the geometric diameter of posts. Since the spacing between posts *S*_p_ used in these simulations is always larger than the interaction cut-off distance $r_{c}=2^{1/6}\sigma$, an effective monomer could potentially interact with maximum of the four nearest posts (Figure 1a). 

The effective bonds were represented by the finitely extensible nonlinear elastic (FENE) potential:(3)$$U_{\mathrm{FENE}}\left(l_{ij}\right)=-\frac{\kappa }{2}R_{o}^{2}\ln\left[1-{\left(\frac{l_{ij}}{R_{o}}\right)}^{2}\right]\;\;\;$$
where $l_{ij}$ is the effective bond length and the FENE parameters $\kappa =30\varepsilon \sigma^{-2}$ and $R_{o}=1.5\sigma$ stand for the spring constant and maximal allowable bond length, respectively. In addition to the FENE potential, the other contribution to the resultant length of effective bond is the WCA potential (Equation (1)). This combination of potentials provides the average bead size *w* ≅ 0.9*σ* and the average effective bond length ⟨l⟩≅0.97σ. With w≅⟨l⟩ this model may be classified as the touching bead model of polymer chains. The effective diameter of a post thus writes *d*_p_ = *D*_p_ + *w*. In this paper, *w* = *l* enters the expressions of investigated quantities and the respective plots. 

The bending potential for the discretized WLC chain was derived from the elastic energy of a WLC in a continuum limit:(4)$$U_{\mathrm{WLC}}=\frac{1}{2}B\int_{0}^{L}ds{\left(\frac{du\left(s\right)}{ds}\right)}^{2}\;$$
where *s* is the arc length, ***u***(*s*) is the corresponding tangent unit vector, and *B* is the elastic constant related to the chain persistence length *P* = *B*/*k*_B_*T*. The discretized version of this WLC energy $\left(B/2l\right)\overset{N-1}{\underset{i=1}{\sum }}{\left(u_{i+1}-u_{i}\right)}^{2}\;$=$\;\left(B/l\right)\overset{N-1}{\underset{i=1}{\sum }}\left(1-u_{i}u_{i+1}\right)$ with ***u****_i_* being the unit vector of *i*-th bond leads to the frequently adopted form of the bending potential energy:(5)$$U_{b}\left(\theta \right)=\frac{B}{l}\left(1+\cos\theta \right)\;\;\;$$
where *θ* is the valence angle between two consecutive bonds in a chain (i.e., the complementary angle of $u_{i}u_{i+1}$). The chain stiffness was quantified by the dimensionless stiffness parameter, *b* = *B*/*lk*_B_*T*. In order to model a DNA chain, the chain stiffness parameter was set to 20 for the investigated semiflexible chain. This value maps the value of persistence length for a DNA molecule in the high ionic strength conditions when the electrostatic interactions along the DNA backbone are screened and *w* ≅ 2.5 nm, *P* ≅ 50 nm, and *b* = *P*/*w* ≅ 50 nm/2.5 nm ≅ 20. The corresponding persistence length determined from the initial decay of orientation correlations $\langle \cos\theta_{ij}\rangle =\exp\left[-\frac{l\left|i-j\right|}{P}\right]$ for a free semiflexible chain is P≅19.7, in good agreement with the predicted value P=b⟨l⟩≅20×0.97≅19.4. Both the flexible and the semiflexible chain contained 1000 effective monomers. Since for the semiflexible chain one effective monomer represents 7.4 base pairs, this length corresponds to approximately 7.4 kbp.

The nanoposts were organized in a parallel way in a square lattice as displayed in Figure 1. The axes of nanoposts were collinear with the *x* coordinate. The post diameter was kept constant at *D*_p_ = 3.0 and thus the effective post diameter was *d*_p_ = 3.9 while the spacing between the post axes was varied in the 4.0–60.0 range. The evolution of the structure of the confined chains with the variation of the geometry of the nanopost array at constant *d*_p_ is compared with the evolution at constant *S*_p_ = 12 with *d*_p_ covering the range 1.9–11.9 and at constant *w*_p_ = 2 with *S*_p_ and *d*_p_ in the intervals 3.9–62.9 and 1.9–60.9, respectively [49,50]. It has been shown that the structural behavior of a chain confined in the array of nanoposts depends on the way in which the geometry of nanopost array is modified. The size of the interstitial volume, *d*_c_ = $\sqrt{2}$*S*_p_ − *d*_p_, is approximated by the size of a quasi-channel (Figure 1a). Table 1 collects the geometrical parameters of nanopost arrays as well as the parameters quantifying the confinement strength, i.e., the geometry ratio $d_{c}/w_{p}$ and the volume fraction of the posts $F=\pi d_{p}^{2}/4S_{p}^{2}$. 

The minimal achievable post separation *S*_p_ = 3.0 is geometrically bounded by the post diameter *D*_p_. This geometry corresponds to the touching posts and a chain is biaxially confined in a single interstitial space of *d*_c_ ≅ 1.24. At this limit of the post separation, the narrow space available to a confined chain within an interstitial volume is supposed to induce straight chain conformation. However, the value of the effective passage aperture width *w*_p_ is virtually zero even at *S*_p_ = 3.9 when a chain of width *w* ≅ 0.9 would have to overcome interaction energy barrier of ~ 1.15 × 10^5^
*k*_B_*T* when translocating through the passage aperture. The same energetic arguments suggest that a fragment of the confined chain may translocate through a passage width into the neighboring interstitial volume when the post separation *S*_p_ exceeds 4.8. When compared to the modification of post array geometry at constant *S*_p_ = 12 and *w*_p_ = 2 [49,50,51], at constant *d*_p_ = 3.9, the geometrical parameters characterizing the confinement strength *d*_c_ and *w*_p_, cover larger range of values, though, the ratio *d*_c_/*w*_p_ and volume fraction of the posts exhibit smaller variations. Although in this study, the chain confinement of nanoposts at constant *d*_p_ is more relaxed than the chain confinement at constant *S*_p_ or *w*_p_, the post spacing in experimental electrophoretic devices is still much larger and accounts to several multiples of the DNA persistence lengths [10,52,53,54,55]. Besides the posts, there were no other restrictions neither in the parallel nor in the perpendicular directions with respect to the post axes. In principle, no periodic boundary conditions were necessary to be imposed in the simulated systems.

In the simulations, the initial conformation of the completely straightened chain backbone was used with two initial orientations. In one orientation, the chain principal axis was aligned with the post axes while in the second orientation, the chain principal axis was orientated perpendicularly to the post axes. The virtually identical structural properties obtained from both initial orientations after equilibration period proved the ergodicity of the system.

All the MD simulations were performed using the DL_Poly Classic package [56]. The systems were simulated in an *NVT* ensemble where the temperature was kept constant at *T* = *ε*/*k*_B_ applying the Nosé-Hoover thermostat with a relaxation time of 0.1*τ*. The time unit was expressed in the simulation units as *τ* = *σ*(*m*_o_/*ε*)^1/2^, with *m*_o_ = 1 being the unit of mass. The time step was set to 0.005*τ*. The leap-frog algorithm was implemented for the numerical integration of the chain trajectory. After a thermal pre-equilibration lasting 10^5^*τ* the system was equilibrated at *T* = 1*ε*/*k*_B_ for 5 × 10^7^*τ*. During this equilibration, the satisfactory oscillations of the potential energy and radius of gyration were achieved. The equilibration was followed by the production phase which lasted for 2 × 10^8^*τ*. For the analyses of the structural quantities, 2 × 10^4^ conformations were considered, the frames were collected every 10,000th step. For each system, the resulting properties were computed as the averages over the all collected frames from three independent simulations. The size of the chains modified with the post array was characterized by the experimentally measurable mean span of the chain along the post axes defined as:(6)$$R_{s}\equiv \langle \max\left(x\right)-\min\left(x\right)\rangle \;$$
where, max(*x*) and min(*x*) are, respectively, the maximal and minimal values of *x* coordinates of a chain in the post array. As the stretching of a linear chain induced by biaxial confinement increases, the mean span approaches the end-to-end distance [35]. The occupation number provided the extent of chain partitioning among the interstitial volumes in the array of nanoposts and expressed the number of interstitial spaces occupied by the monomers. The internal organization of monomers within a chain was studied using the single chain structure factor. The free linear chain composed of 1000 effective beads was considered as a reference.

## 3. Results and Discussion

### 3.1. Occupation Number

The occupation number, i.e., the number of interstitial volumes occupied by a chain is plotted as a function of the geometry ratio *d*_c_/*w*_p_ and volume fraction *F*, respectively, for all three types of geometry modification in Figure 2a,b. One can see that the trend of the occupation number depends on the modification type. The observed trends in the evolution of occupation number with the geometry of the nanopost array may be rationalized using the free energy arguments and the geometrical approximations for the space restricted by the posts. Specifically, an interstitial volume may be well approximated by the channel geometry of size *d*_c_ while the narrow aperture may be assumed as a slit of height *w*_p_ [47,48,49,50,51]. The distribution of a chain in an array of posts is determined by the structural parameters of the chain and by the geometrical parameters of the post array. In the equilibrium, the free energy of chain fragments in the interstitial volumes should be equal to the free energy of chain fragments situated in the passages between two adjacent posts. Depending on the size of interstitial volumes and passage apertures, the chain fragments in a quasi-channel or quasi-slit might adopt conformation characteristic for either the de Gennes regime [57,58] or the Odijk regimes [28]. In the de Gennes regime, the free energy excess of a chain confined in a channel of square cross-section reads [20]:(7)$$\Delta A/k_{B}T=4.0L{\left(Pw\right)}^{1/3}D^{-5/3}$$
where *w* is the chain width, *L* is the chain contour length, and *D* is the size of the channel approximated by *d*_c_ in the nanopost array. The free energy of a semiflexible chain in the strong confinement of a square channel (Odijk regime) is given by [34,59]:(8)$$\Delta A/k_{B}T=2.2072LP^{-1/3}D^{-2/3}$$

It has been shown that the free energy of a chain confined in a slit is, to a good approximation, half the free energy of a chain confined in a square channel of the same side as the height of a slit [60]. Thus, Equations (7) and (8) divided by a factor of 2 may be used to calculate the free energy of a chain in a slit under moderate confinement and of a semiflexible chain in a slit under strong confinement, respectively. (1.) Considering this relation and the equality between the free energy relations (7) or (8) for a chain in a square channel (substituting *d*_c_ for *D*) or in a slit (substituting *w*_p_ for *D*), respectively, and (2.) substituting the fraction of chain length in interstitial volumes, *L*_qc_, and in passage apertures, *L*_qs_, for *L* = *L*_qc_ + *L*_qs_ in Equations (7) and (8), respectively, provide the dependence of the chain length distribution on the *d*_c_/*w*_p_ ratio in a nanopost array. This dependence is presented for all three combinations of confinement regimes in Table 2 and controls the occupation number. It is apparent that the ratio *L*_qc_/*L*_qs_ is a function of the chain parameters, i.e., the persistence length and width of the chain only if the conformation of a chain fragment in interstitial volumes is governed by the de Gennes regime while the conformation of a chain fragment in passage apertures is governed by the Odijk regime. The numerical constants in Equations (7) and (8) and, consequently, in Table 2 should be not considered strictly for the investigated systems, since a square channel and a slit are only geometrical approximations to the geometry of the interstitial volume and the passage aperture, respectively.

As can be seen in Figure 2, at constant *w*_p_ = 2 and *S*_p_ = 12, the occupation number of the flexible and semiflexible chain monotonically decreases with the increasing *d*_c_/*w*_p_ ratio or volume fraction (increasing *d*_p_). Although, the passage aperture is wide enough for the chain translocation in the case of constant *w*_p_ = 2, the occupancy of a single interstitial volume is found for the flexible chain if *d*_c_/*w*_p_ ≥ 7.8 and for the semiflexible chain if *d*_c_/*w*_p_ ≥ 14.0. In the case of constant *S*_p_ = 12, for both the flexible and the semiflexible chain, the occupancy of a single interstitial volume is observed for *d*_c_/*w*_p_ ≥ 5.5, which corresponds to *w*_p_ = 1.1. At lower *d*_c_/*w*_p_, the relative free energy penalty of a chain in an interstitial volume becomes high enough for a chain to expand from a single interstitial volume to more interstitial volumes through slit-like apertures. In comparison with the flexible chain, the semiflaxible chain experiences larger restriction in a channel-like interstitial volume of *d*_c_ > *P* (Equation (7)) and lower free energy barrier in the slit-like aperture (Equation (8)) which is the reason why it starts to expand into more interstitial volumes already in nanopost arrays of larger *d*_c_/*w*_p_. It can be inferred from the higher values of the occupation number for the semiflexible chain than for the flexible chain that the persistence length plays a role in the chain distributions. While the de Gennes regime controls the conformation of the flexible chain in the interstitial spaces as well as in the passage aperture (entry 1 in Table 2), the conformation of the semiflexible chain is supposed to be controlled by the Odijk regime in the passage aperture (entries 2 or 3 in Table 2).

At constant *d*_p_ = 3.9, the evolution of the occupation number displays a maximum with approximately the same position at *d*_c_/*w*_p_ ≅ 1.9 for both, the semiflexible and the flexible chain. In this geometry arrangement, the small values of *d*_c_/*w*_p_ ratio correspond to wide interstitial volumes and wide passage aperture as can be seen in Table 3, where these values are compared with the geometrical parameters associated with the geometry modifications of post array at constant *w*_p_ = 2 and *S*_p_ = 12. Thus, at small *d*_c_/*w*_p_ ratio, the chains are only very modestly confined. This might be deduced from the comparison of the radius of gyration (*R*_g_) of chains at this confinement with the radius of gyration of free analogues. For both the flexible and the semiflexible chain, *R*_g_ of confined chains only slightly exceeds *R*_g_ of free chains (29.9 vs. 27.3 for *b* = 0 and 75.5 vs. 68.4 for *b* = 20). Moreover, the axial and lateral components of *R*_g_ for confined chains are identical within the standard deviation and their values are ~ *R*_g_/$\sqrt{3}$. The average occupation number of the flexible chain in this post array is less than 2 and of the semiflexible chain only slightly more than 5. On the other hand, at constant *w*_p_ = 2 or at constant *S*_p_ = 12, the larger values of *d*_c_/*w*_p_ correspond to narrow interstitial volume and narrow passage aperture and, in order to release the constraints, the chain passes through the apertures and penetrates into more interstitial volumes. That is why at small *d*_c_/*w*_p_ the chain occupies more interstitial volumes than in the case of constant *d*_p_. It follows that the occupation number of the chain in the nanopost array of constant *d*_p_ = 3.9 depends rather on *R*_g_ of only slightly restricted chain, whose size is almost unmodified by the presence of posts. Decrease of the post separation (increase of *d*_c_/*w*_p_) enhances crowding of monomers in the nanopost array, which is released when monomers penetrate through the passage apertures into more interstitial volumes. This explains the initial increase of the occupation number up to *d*_c_/*w*_p_ ≅ 1.9 for both the flexible and the semiflexible chain (Figure 2). With the further increase of *d*_c_/*w*_p_ ratio, the occupation number starts to decline, which indicates that the confinement of the chain in an interstitial volume becomes attenuated when compared with the confinement in a passage aperture and longer chain fragments are stored in lower number of interstitial volumes. The uniform position of the maximum for the flexible and semflexible chain indicates that the chain conformations in the interstitial volumes as well as in the passage spaces are governed by the same conformation regime in the post arrays of the geometrical parameters where the maximum is located. Consulting Table 2, it appears that in the interstitial volumes as well as in the passage apertures, the conformation of the flexible chain corresponds to de Gennes regime (entry 1 in Table 2) while the conformation of the semiflexible chain corresponds to Odijk regime (entry 3 in Table 2). The decay of the occupation number at larger *d*_c_/*w*_p_ ratio originates from the more significant free energy excess of the chain fragments in the passage aperture than in the interstitial volumes as has been already discussed for the geometries with constant *S*_p_ and *w*_p_. The occupancy of a single interstitial volume starts at *d*_c_/*w*_p_ ≥ 4.1 (corresponding to *w*_p_ = 0.6) for both the flexible and semiflexible chain.

The occupation number of the semiflexible chain is higher than the occupation number of the flexible chain independent of the variation of the geometry of the nanopost array. The enhanced tendency of stiffer chains to expand into more interstitial volumes has been already rationalized in previous studies [48,49,50,51]. The frequency of collisions between a chain and the walls of posts during the translocation through a passage aperture is reduced for stiffer chains and thus they penetrate through the passage aperture more easily. The more quantitative explanation is based on the slit-like approximation for the narrow passage aperture which imposes a strong confinement on a translocating chain. Since the persistence length of a chain exceeds the passage width, the chain conformations belong to the Odijk regime where the free energy of a chain scales inversely as ~ *P*^−1/3^ with the chain persistence length. Thus, the stiffer chain overcomes lower free energy barrier during the translocation through a passage aperture. Moreover, if the chain fragments obey the statistics of the de Gennes regime in interstitial volumes, where the free energy of the confined fragments increases with ~ *P*^1/3^, the stiffer chain tends to reduce the number of monomers in individual interstitial volumes and the occupation number increases. At sufficiently large values of *d*_c_/*w*_p_ (or *F*), a single occupancy of both chains is observed for all three geometry modifications. As this ratio decreases the difference in the free energy excess of chain fragments in interstitial volumes and passage apertures decreases and the chain fragments penetrate through the apertures into the neighboring interstitial volumes.

### 3.2. Chain Extension Along the Post Axes

The chain extension along the post axes defined as the span in Equation (6) is presented as a function of the *d*_c_/*w*_p_ ratio and volume fraction of posts in Figure 3a,b for the flexible and semiflexible chain and different geometry modifications of the post array. At constant *d*_p_ or *S*_p_, the span monotonically increases with the confinement strength (increasing *d*_c_/*w*_p_ ratio or volume fraction of posts) while at constant *w*_p_, a maximum arises. As can be seen in Table 3 and Figure 2, the increase of *d*_c_/*w*_p_ is accompanied by the decrease of the size of the interstitial volumes and occupation number when *S*_p_ is kept constant. These both factors increase the crowding of monomers in the individual interstitial volumes, which results in the axial chain extension. The trend of the axial extension of a chain in the series of nanopost array with constant *d*_p_ monotonously increases despite the presence of a maximum in the evolution of the occupation number with the *d*_c_/*w*_p_ ratio. This implies that at small *d*_c_/*w*_p_, the effect arising from the reduced width of interstitial volumes dominates.

However, at constant *w*_p_, the increase of *d*_c_/*w*_p_ ratio or volume fraction of posts is associated with the increase of the size of the interstitial volumes and decrease of the occupation number. These two factors influence the axial chain extension in the opposite way. Upon the decrease of the occupation number, more monomers are located in the individual interstitial volumes promoting the axial chain extension. However, at the same time, the size of the interstitial volumes increases which releases the lateral restriction of monomers in individual interstitial volumes and is responsible for the decrease of the axial chain extension. At lower values of *d*_c_/*w*_p_, the increasing trend of the axial chain extension indicates that the effect of the axial chain extension due to the increased crowding of monomers prevails. On the other hand, at higher values of *d*_c_/*w*_p_, when the occupation number levels off, the decreasing trend of the axial chain extension indicates that the lateral release of chain fragments in individual interstitial volumes dominate the overall axial chain extension. The position of the maximum in the axial chain extension is situated at lower values of *d*_c_/*w*_p_ for the flexible chain than for the semiflexible chain (*d*_c_/*w*_p_ ≅ 3.7 vs. 5.7) which is attributed to larger occupation number of the semiflexible chain and larger values of *d*_c_/*w*_p_ at which the semiflexible chain achieves a single occupancy.

At large values of *d*_c_/*w*_p_ or *F*, when the single occupancy dominates, the axial chain extension is most significant for the geometry modification at constant *d*_p_ while the smallest axial chain extension is found for the geometry modification at constant *w*_p_. Since at single occupancy, the axial chain extension is solely dictated by the size of the interstitial space, this order well correlates with the width of the interstitial space *d*_c_, which is the smallest for the geometry modification at constant *d*_p_ and largest for the geometry modification at constant *w*_p_.

In spite of the larger occupation number, the semiflexible chain exhibits larger axial extension than the flexible chain because of the larger axial extension of stiffer chain fragments in individual interstitial volumes. This effect is least significant for the geometry modification at constant *d*_p_.

In order to check whether the approximation of the interstitial space by a channel is adequate for the geometry variation at constant post diameter, the axial chain extension in logarithmic scale is plotted against the size of the interstitial space in Figure 4. The adequacy of such an approximation has been already validated for the geometry variation of the post array at constant *S*_p_ and *w*_p_ [49]. In narrow channels with the size of cross-section *D* < *P*, the semiflexible chain is in the Odijk regime [28] and its longitudinal extension is given by:(9)$$R=L\left[1-A{\left(\frac{D}{P}\right)}^{2/3}\right]$$
where the numeric constant *A* is 0.1701 and 0.18274 for the channel of a circular and square cross-section, respectively [34,59]. At moderate confinement, i.e., in larger interstitial space, when the chain is either in the extended de Gennes regime (*P* < *D* < *P*^2^/*w*) [35,61,62] or in the classic de Gennes regime (*D* > *P*^2^/*w*) [57,58], the axial chain extension is defined as:(10)$$R\approx L{\left(\frac{wP}{D^{2}}\right)}^{\frac{1}{3}}$$

At single occupancy, where the channel-like confinement might be expected, the interstitial space is rather narrow, *d*_c_ < *P*. It implies that the semiflexible chain is expected to be in the Odijk regime. Since the Odijk regime is distinctive only for semiflexible chains, the flexible chain is supposed to be in the de Gennes regime. In Figure 4, one can see that the axial chain extension at single occupancy satisfactorily follows the Odijk regime for the semiflexible chain and there is also an indication of the power low dependence of the axial chain extension for the flexible chain. The trend of the longitudinal component of the radius of gyration (parallel to the post axes) as well as the trend of the overall radius of gyration followed the trend of the axial chain extension. The trend of the lateral component of the radius of gyration (perpendicular to the post axes) followed the trend of the occupation number.

### 3.3. Structure Factor

The single-chain structure factor provides important information on the statistical organization of chain segments. The hierarchy of the chain organization on different length scales can be studied when this quantity is plotted as a function of the wavevector *q* = 2*π/Ω*. Here, *Ω* represents the length scale ranging from the monomer size up to about the chain size. The single-chain structure factor is defined as follows:(11)$$S\left(q\right)=\frac{1}{N^{2}}\langle \overset{N}{\underset{i=1}{\sum }}\overset{N}{\underset{j=1}{\sum }}\sin\left(qr_{ij}\right)/qr_{ij}\rangle \;$$
where *r**_ij_* is the distance between segments *i* and *j*. The information of the chain organization is stored at moderate values of the wavevector, *q* > 2*π*/*R*_s_ where *S*(*q*) ~ *q*^−1/*ν*^ (*ν* is the Flory scaling exponent). At smaller values of wavevectors, where the structure factor saturates, only the information on the chain size can be extracted. The onset of saturation is shifted to lower values of *q* for more extended chains. In the relevant range, the logarithmic plot of *S*(*q*) of biaxially confined chain might display several salient regimes [63,64,65,66].

The structure factor of a semiflexible chain confined in a symmetric channel, which is assumed as an approximation to the geometry of the interstitial space, contains information on organization of monomers at different length scales [63]. The persistence length scale is characterized by the slope −1 for *q* > 2π/*P*. Under moderate confinement, when a chain is viewed as a linear array of statistical blobs, the slopes −5/3 or −2 might be expected. The slope −5/3 points to the classic de Gennes regime for the channel diameters *D* > *P*^2^/*w* and its interval is delimitated by 2*π*/*P* > *q* > 2*π*/*D*. The slope −2 points to the extended de Gennes regime for the channel diameters *P* < *D* < *P*^2^/*w* and is delimitated by 2*π*/*P* > *q* > 2*π*/(*D*^2^*P*^2^/*w*)^1/3^. For short biaxially confined chains, the slope −2 might also signify the pseudoideal regime. At 2*π*/*R*_s_ < *q* < 2*π*/*D* (classic de Gennes regime) or at 2*π*/*R*_s_ < *q* < 2*π*/(*D*^2^*P*^2^/*w*)^1/3^ (extended de Gennes regime), the slope −1 signifies the linear arrangement of blobs under moderate confinement of a symmetrical channel. The Odijk regime provides very simple structure factor with the slope −1 over all wavevectors *q* > 2*π*/*R*_s_. The appearance of all slopes in one plot of *S*(*q*) function requires a sufficiently long chain with a sufficient number of the persistence lengths confined in a channel which is broad enough for the blobs to be developed.

The structure factor of a biaxially confined flexible chain is supposed to be simpler. The slope −1 is expected at *q* > 2*π*/*l* (*l* is the effective bond length) and at 2*π*/*R*_s_ < *q* < 2*π*/*D*. At 2*π*/*l* > *q* > 2*π*/*D*, the slope –5/3 characteristic for the de Gennes regime or the slope −2 characteristic for the pseudoideal regime may be found.

The structure factor in the logarithmic scale for the flexible chain confined in the array of nanoposts of constant *d*_p_ at various post separations along with the structure factor of the free flexible chain are presented in Figure 5a. 

At small post separation of *S*_p_ = 4, the chain occupies only a single interstitial space which is narrow and thus no three-dimensional blob regime can be observed. The chain is extended along the post axis, which is reflected on the slope –1 over the all interval of *q* values. In the nanopost array with the post separation *S*_p_ = 5, the intensity of *S*(*q*) is slightly enhanced which reflects the monomer correlations also in the lateral direction with respect to the post axis. This correlation happens within a single interstitial space (*n*_p_ = 1). The characteristic feature of a chain occupying more interstitial volumes in the post array is the presence of a hump in the structure factor, which is detected for the post arrays with *S*_p_ = 6, 7, 9, and 11 and shown in Figure 5b. The position of this hump well correlates with the post separation as shown in Table 4 where 2*π*/*q* values are in good agreement with the corresponding *S*_p_ values. Thus, the presence of a hump indicates enhanced correlations of monomers contained in the two chain fragments, which are in mutual parallel arrangement separated by one post. The analogous correlations between monomers of two chain fragments separated by two or more nanoposts expected at *q* ≅ 2*π*/2*S*_p_, 2*π*/3*S*_p_… have not been observed in the structure factor. The appearance of these additional humps, most likely, requires a longer chain. The hump disappears at larger post separations when the structure factor of the chain virtually coincides with the structure factor of the corresponding free chain (Figure 5a). The statistics of monomer organization on the length scale smaller than the post separation demonstrates itself at *q* > 2*π*/*S*_p_, while information about the statistics of monomer organization on the length scale larger than the post separation can be obtained at *q* < 2*π*/*S*_p_. For *S*_p_ ≥ 14, the structure factor of the flexible confined chain is essentially identical with the structure factor of the free flexible chain. It thus appears that the presence of sparsely distributed narrow posts only slightly affects the statistics of a flexible chain on the length scale larger than post separation. Such a behavior has been already observed for a flexible and semiflexible chain confined in the nanopost array of *d*_p_ = 1.9 and *S*_p_ = 12 [49]. This finding also supports the above explanation of the maximum observed in the occupation number plotted against *d*_c_/*w*_p_ or *F* (Figure 2).

Figure 6 shows the logarithmic plot of the structure factor for the semiflexible chain confined in the array of nanoposts of constant *d*_p_ at various post separations. The structure factor for the free semiflexible chain is also included for comparative purposes. Again, at large *d*_c_/*w*_p_ ratio, the single occupancy of the chain is reflected on slope −1 over the all investigated interval of wavevectors. In contrast with the flexible chain, there is no hump characterizing the post separation in *S*(*q*) for the semiflexible chain in the post arrays with *n*_p_ > 1. 

The larger stiffness of the chain backbone prevents the chain fragments separated by a post from adopting mutually parallel arrangements of the U-shape in a statistically relevant number. In a similar way to the structure factor of the flexible chain, the structure factor of the semiflexible chain confined in the post array with larger post separation is very similar to the structure factor of the semiflexible free chain. For *S*_p_ = 14, the structure factor of the confined chain is essentially identical with the structure factor of the free chain. The statistics within one blob of a chain fragment formed by the interstitial volumes, however, is not observed in the structure factor of the semiflexible chain. The appearance of the classic de Gennes regime (slope −5/3) requires the size of biaxial confinement at least of the order of ~ *P*^2^/*w* = 430, which is much larger than the size of the interstitial volumes, *d*_c_, of the investigated nanopost arrays. Thus, the extended rather than the classic de Gennes regime might be expected. At larger *S*_p_ values, the number of asymmetric blobs formed in the extended de Gennes regime (slope −2) is of the order of $L{\left(w^{2}/d_{c}^{4}P\right)}^{1/3},$ which is even smaller than the occupation number of the chain (the number of occupied quasi-channels). Thus, there are not enough asymmetric blobs for the statistics of the extended de Gennes regime to be observed in individual interstitial volumes. More than one asymmetric blob could be formed at smaller *S*_p_ but the narrow interstitial space gives rise to the Odijk regime (Figure 4). In the structure factor of semiflexible chain, the slope −1 crosses directly to the saturated region at smaller wavevectors.

## 4. Conclusions

Coarse-grained molecular dynamics simulations have been used to study the structural behavior of flexible and semiflexible chains confined in an array of nanoposts collinearly organized in a square lattice. The geometry of the post array has been varied at constant post diameter *d*_p_ = 3.9 and the results obtained for both chains in this series of nanopost arrays have been compared with the results obtained for both chains confined in nanopost arrays where the geometry has been varied at constant post separation *S*_p_ = 12 or at constant passage width *w*_p_ = 2. The findings have been rationalized in terms of approximation of the interstitial space and passage aperture to the channel and slit geometry, respectively. Since the distribution of monomers is a compromise between the interstitial volumes (defined by the diameter of the quasi-channel, *d*_c_) and the passage apertures of the width *w*_p_, the geometry of the nanopost array has been defined using the confining ratio *d*_c_/*w*_p_ and the volume fraction *F*.

While the occupation number monotonically decreases with increasing *d*_c_/*w*_p_ ratio and volume fraction at constant *w*_p_ and *S*_p_ regardless of the chain stiffness it achieves a maximum positioned at *d*_c_/*w*_p_ ≅ 1.9 for both, the flexile and the semiflexible chain at constant *d*_p_. The increasing trend of the occupation number at lower *d*_c_/*w*_p_ for the series of nanopost arrays with constant *d*_p_ follows from the combination of wide interstitial volumes and wide passage apertures. The conformation of the chain is only slightly modified in these confining geometries and the initial increased penetration upon the increase of *d*_c_/*w*_p_ ratio is associated with the effort of the chain to release the increasing constraints via its redistribution into more interstitial volumes. The uniform position of the maximum for both the flexible and semiflexible chain points out to the same confinement regime in the interstitial volumes and in the passage aperture. However, the different height of this maximum for both chains indicate different conformation regimes. While in the case of a flexible chain, the de Gennes regime governs the statistics of monomer organization, in the case of a semiflexible chain, the Odijk regime is supposed to control the chain organization. The decrease of the occupation number with the *d*_c_/*w*_p_ ratio is explainable by free energy arguments applied to the channel- and slit-like approximations of the interstitial volumes and passage apertures, respectively. In order to achieve a balance of the free energy excess with increasing *d*_c_/*w*_p_ in both types of confinement, the monomers concentrate in relatively broader interstitial volumes rather than in relatively narrower passage aperture. Because of the higher free energy penalty of the stiffer chain fragments in the interstitial volumes in the de Gennes regime and lower free energy barrier to the translocation of the stiffer chain fragments through the passage aperture the occupation number of the semiflexible chain is higher than the occupation number of the flexible chain in all three investigated series of nanopost arrays. Because of the lower free energy barrier to the translocation of the stiffer chain fragments through the passage aperture the semiflexible chain starts to expand from the single interstitial volume into more interstitial volumes already at larger values of *d*_c_/*w*_p_ when compared with the flexible chain at constant *w*_p_.

The axial chain extension is a monotonically increasing function of the *d*_c_/*w*_p_ ratio or volume fraction when the geometry of the nanopost array is varied at constant *d*_p_ or *S*_p_ whereas maximum is found in this plot for both the flexible and semiflexible chain when the geometry of nanopost array is varied at constant *w*_p_. The increase of *d*_c_/*w*_p_ ratio at constant *S*_p_ is accompanied by the decrease of the size of interstitial volumes, *d*_c_, and by the increase of the number of monomers within the individual interstitial volumes. These both effects promote the axial chain extension. The increase of *d*_c_/*w*_p_ ratio at constant *w*_p_ leads to the increase of the size of interstitial volumes and to the increase of the number of monomers within the individual interstitial volumes. While the former effect acts against the axial chain extension, the latter effect is responsible for the axial chain expansion within the individual interstitial volumes. The existence of the maximum implies that at smaller *d*_c_/*w*_p_ values, the effect of the increased crowding of monomers prevails and at larger *d*_c_/*w*_p_ values, the effect of the geometric lateral relaxation in interstitial volumes dominates. At constant *d*_p_, the axial extension induced by the decreasing size of interstitial volume prevails over the decreased number of monomers within interstitial volumes at small *d*_c_/*w*_p_ values.

The single structure factor plotted as a function of the wavevector in logarithmic scale is used to reveal the organization of monomers at different length scales and to find out how the geometry of nanopost array affects the mutual monomer correlations. The structure factor of the flexible chain in the presence of nanoposts contains more information when compared with the structure factor of the semiflexible chain since the chains are not too long and the confining space is not too large for all predicted hierarchical organizations of the semiflexible chain to be developed. At small post separations, *S*_p_ ≤ 5, the structure factor of the flexible chain resembles the structure factor of a strongly biaxally confined chain. The geometry of the nanopost array reflected on the presence of a hump demonstrates itself only for *S*_p_ = 6, 7, 9, 11 and its position displays good correlation with the respective post separation. At larger *S*_p_ (small *d*_c_/*w*_p_), the relaxed conformation of the flexible chain in the array of more sparsely distributed nanoposts provides structure factor virtually identical with the structure factor of an analogous free chain. The geometry of the nanopost array is not identified in the structure factor of the semiflexible chain because the chain is too stiff to form a statistically relevant number of U-turns around a post and too short to turn around more than one post. Theoretical analysis rules out the existence of the de Gennes regime however, the small number of asymmetric blobs prevents drawing some firm conclusions on the organization of monomers in the chain fragments within individual interstitial volumes. In the case of narrow interstitial volumes with single chain occupancy (small post separations), the structure factor possesses slope −1 over the all range of wavevectors in agreement with the predictions for a semiflexible chain confined in a narrow channel (Odijk regime).

This study completes the investigation of the effects arising from the different ways of geometry variation of the nanopost array on the structure of a confined flexible and semiflexible chain. The free energy arguments suggest that if the same confinement regime governs the conformation of a confined semiflexible chain the occupation number is supposed to be independent of the persistence length. In order to obtain more comprehensive insight into the effect arising from the chain stiffness, chains of different stiffness parameters need to be studied. Combination of narrow posts with large separations among them affects the conformation of the investigated confined semiflexible and flexible chain only negligibly.

## Figures and Tables

**Figure 1 polymers-12-01064-f001:**
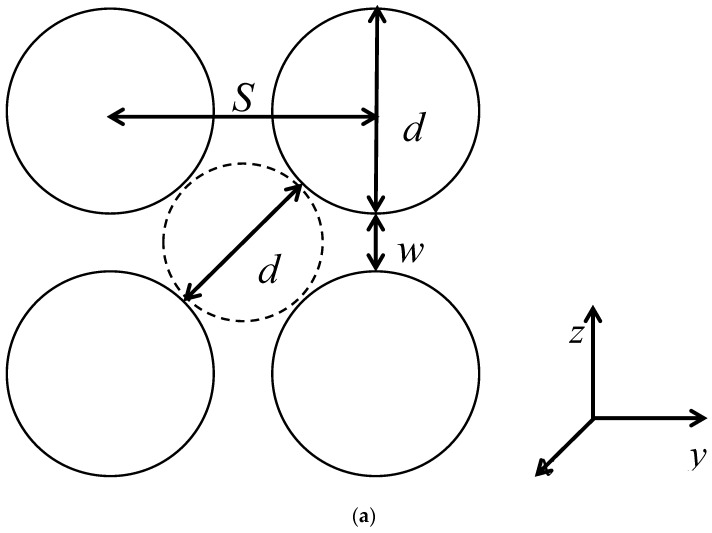
Definition of the parameters characterizing the square-lattice arrangement of the nanoposts illustrated in the cross-sectional view, the Cartesian coordinate frame depicts the orientation of nanoposts. *S*_p_ is the separation of two adjacent nanoposts, *d*_p_ is the effective diameter of nanoposts, *w*_p_ = *S*_p_ − *d*_p_ is the closest distance between the walls of two adjacent nanoposts, i.e., the passage width, and *d*_c_ = $\sqrt{2}$*S*_p_ − *d*_p_ is the diameter of a quasi-channel, which serves as a gauge for the size of the interstitial space embraced by four neighboring nanoposts (**a**). Cross-sectional view of a flexible chain confined in the nanopost array of *S*_p_ = 12 (**b**).

**Figure 2 polymers-12-01064-f002:**
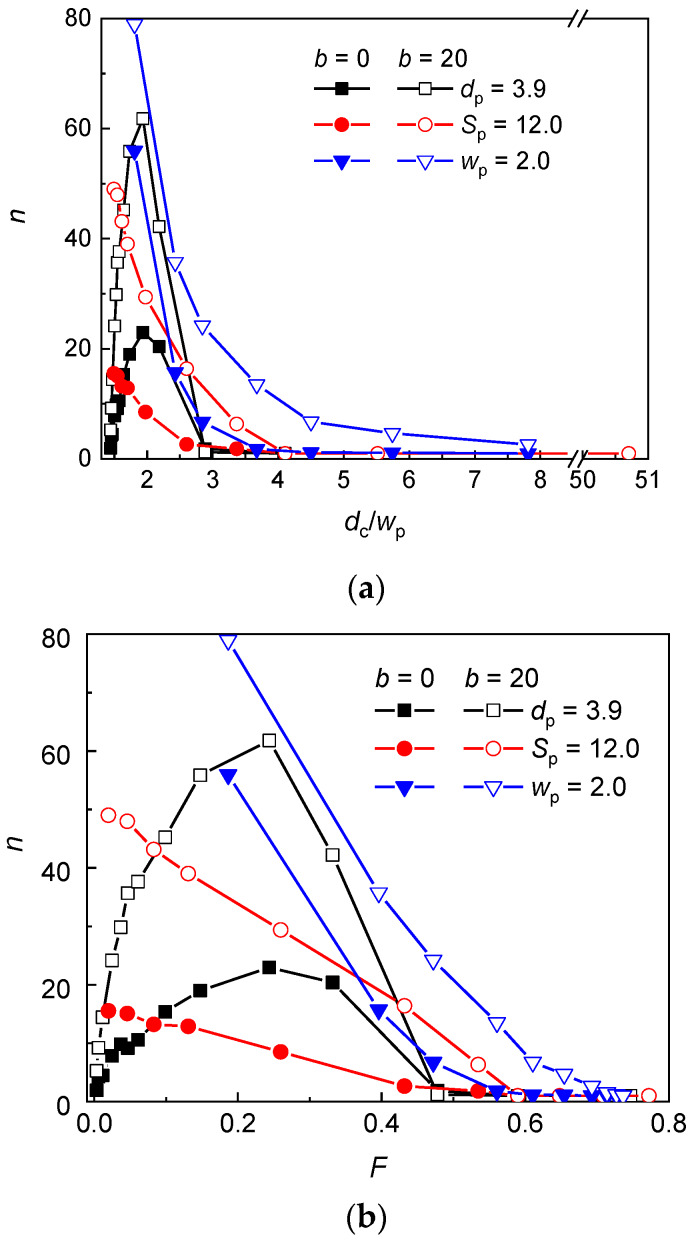
Occupation number of a flexible (*b* = 0) and semiflexible (*b* = 20) chain in a post array with the geometry variation at constant post diameter *d*_p_, post separation *S*_p_, and passage width *w*_p_ as a function of the *d*_c_/*w*_p_ ratio (**a**) and the volume fraction of posts *F* (**b**).

**Figure 3 polymers-12-01064-f003:**
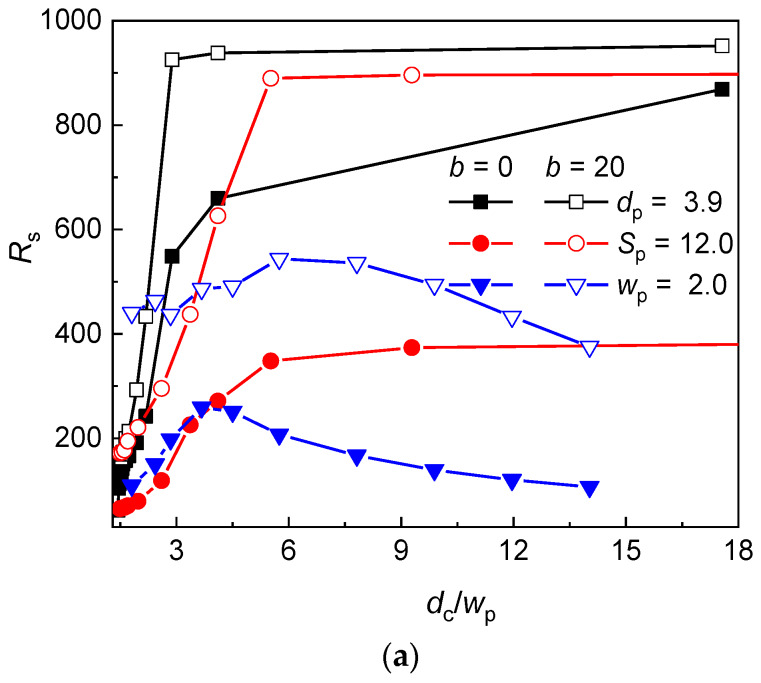
Axial chain extension of a flexible (*b* = 0) and semiflexible (*b* = 20) chain in a post array with the geometry variation at constant post diameter *d*_p_, post separation *S*_p_, and passage width *w*_p_ as a function of the *d*_c_/*w*_p_ ratio (**a**) and the volume fraction of posts *F* (**b**).

**Figure 4 polymers-12-01064-f004:**
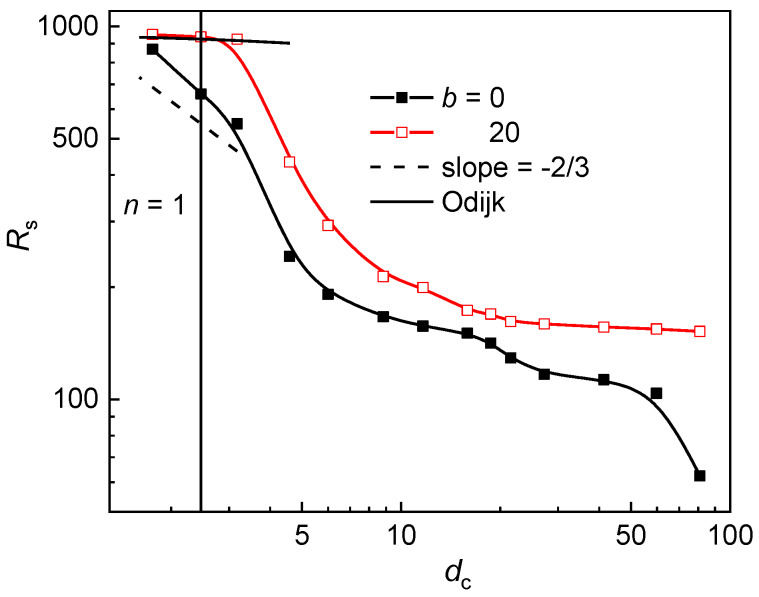
Axial chain extension of a flexible (*b* = 0) and semiflexible (*b* = 20) chain in a post array with the geometry variation at constant post diameter *d*_p_ as a function of the size of the interstitial volume *d*_c_ in a logarithm scale. The solid vertical line demarcates the region with the dominance of single-occupancy for both chains.

**Figure 5 polymers-12-01064-f005:**
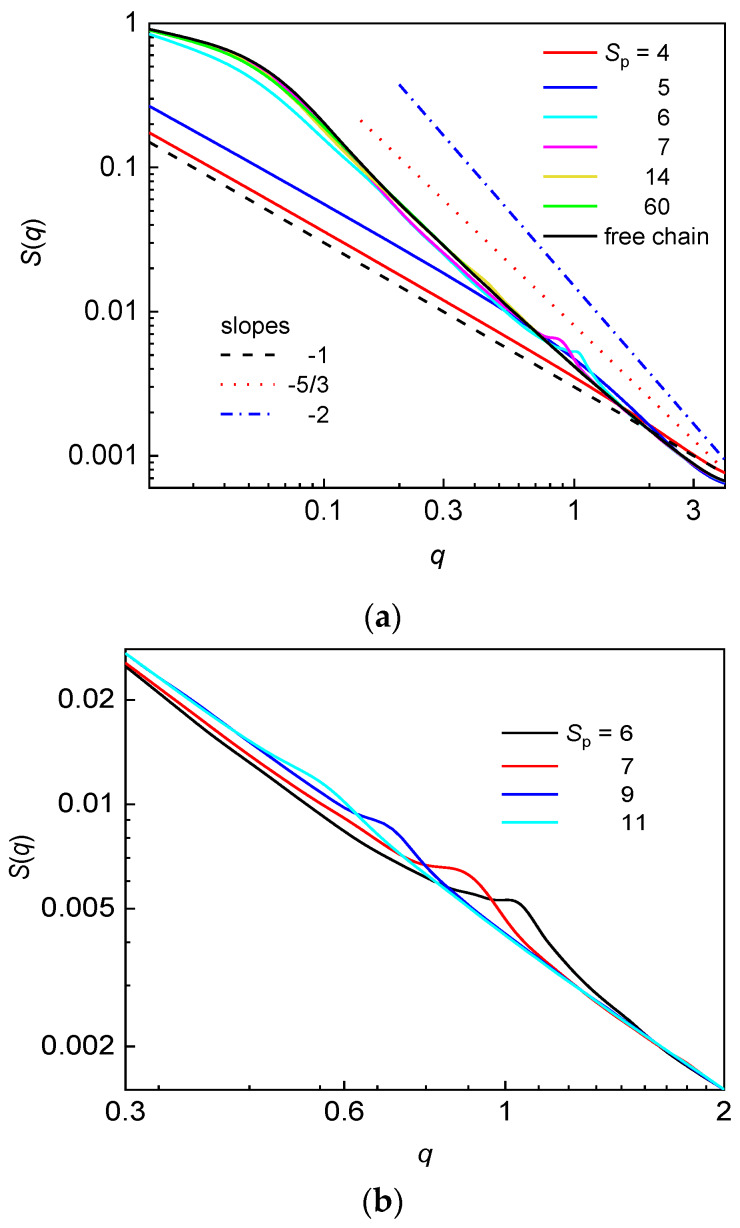
Structure factor for a flexible chain confined in an array of nanoposts of fixed post diameter *d*_p_ = 3.9 and various post separations *S*_p_ (**a**). The straight lines represent the characteristic slopes that might be expected for a flexible chain confined in a symmetric channel. Structure factor for a flexible chain in the interval of wavevectors *q*, where the hump characterizing the geometry of post array is developed (**b**).

**Figure 6 polymers-12-01064-f006:**
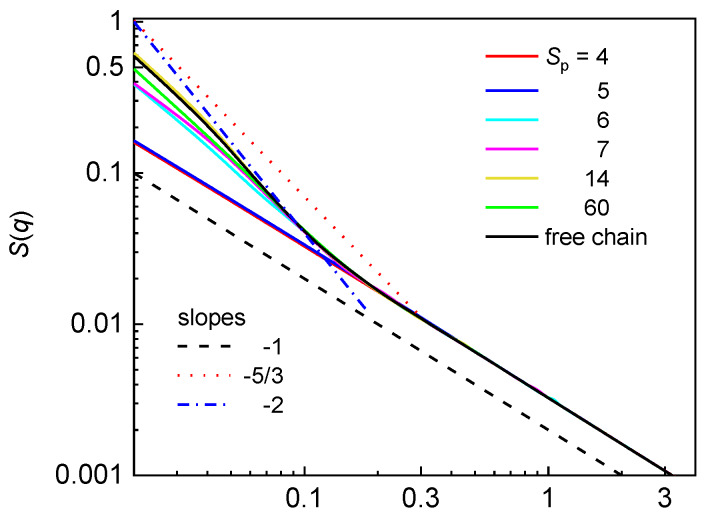
Structure factor for a semiflexible chain confined in an array of nanoposts of fixed post diameter *d*_p_ = 3.9 and various post separations *S*_p_. The solid lines represent the characteristic slopes that might be expected for a semiflexible chain confined in a symmetric channel.

**Table 1 polymers-12-01064-t001:** Geometrical parameters of post arrays with constant post diameter *d*_p_ defined in the text; 1 length unit ≈ 2.5 nm.

*d*_p_ = 3.9									
*S* _p_	*w* _p_	*d* _c_	*d*_c_/*w*_p_	*F*	*S* _p_	*w* _p_	*d* _c_	*d*_c_/*w*_p_	*F*
4.0	0.1	1.757	17.569	0.747	14.0	10.1	15.899	1.574	0.061
4.5	0.6	2.464	4.107	0.590	16.0	12.1	18.727	1.548	0.047
5.0	1.1	3.171	2.883	0.478	18.0	14.1	21.556	1.529	0.037
6.0	2.1	4.585	2.183	0.332	22.0	18.1	27.213	1.503	0.025
7.0	3.1	5.999	1.935	0.244	32.0	28.1	41.355	1.472	0.012
9.0	5.1	8.828	1.731	0.147	45.0	41.1	59.740	1.454	0.006
11.0	7.1	11.656	1.642	0.099	60.0	56.1	80.953	1.443	0.003

**Table 2 polymers-12-01064-t002:** Dependence of the ratio *L*_qc_/*L*_qs_ of the chain length situated in interstitial volumes and passage apertures, *L*_qc_ and *L*_qs_, respectively, on *d*_c_/*w*_p_ ratio in the three possible combinations of confinement regimes.

Interstitial Volume	Passage Aperture	*L*_qc_/*L*_qs_
de Gennes regime	de Gennes regime	$2^{-1}{\left(d_{c}/w_{p}\right)}^{5/3}$
de Gennes regime	Odijk regime	$0.2759{\left(w_{p}/P\right)}^{2/3}{\left(w_{p}/w\right)}^{1/3}{\left(d_{c}/w_{p}\right)}^{5/3}$
Odijk regime	Odijk regime	$2^{-1}{\left(d_{c}/w_{p}\right)}^{2/3}$

**Table 3 polymers-12-01064-t003:** Values of *d*_c_/*w*_p_ ratio and corresponding passage width *w*_p_ and diameter of the interstitial volume *d*_c_ for post arrays of different geometry modifications defined in the text, the length dimensions are in the simulation units; 1 unit ≈ 2.5 nm.

*d*_p_ = 3.9			*S*_p_ = 12			*w*_p_ = 2		
*d*_c_/*w*_p_	*w* _p_	*d* _c_	*d*_c_/*w*_p_	*w* _p_	*d* _c_	*d*_c_/*w*_p_	*w* _p_	*d* _c_
1.443	56.1	80.953	1.492	10.1	15.071	1.808	2.0	3.615
1.454	41.1	59.740	1.546	9.1	14.071	2.429	2.0	4.858
1.472	28.1	41.355	1.614	8.1	13.071	2.843	2.0	5.687
1.504	18.1	27.213	1.700	7.1	12.071	3.672	2.0	7.343
1.529	14.1	21.556	1.975	5.1	10.071	4.500	2.0	9.000
1.548	12.1	18.727	2.603	3.1	8.071	5.743	2.0	11.485
1.574	10.1	15.899	3.367	2.1	7.071	7.814	2.0	15.628
1.642	7.1	11.656	4.107	1.6	6.571	9.885	2.0	19.770
1.731	5.1	8.828	5.519	1.1	6.071	11.956	2.0	23.912
1.935	3.1	5.999	9.284	0.6	5.571	14.027	2.0	28.054
2.183	2.1	4.585	50.706	0.1	5.071			
2.882	1.1	3.171						
4.107	0.6	2.464						
17.569	0.1	1.757						

**Table 4 polymers-12-01064-t004:** Values of wavevector *q*, at which the hump is observed and corresponding 2*π*/*q* values for different post separations *S*_p_.

*S* _p_	6	7	9	11
*q*	1.02	0.86	0.68	0.56
2*π*/*q*	6.16	7.31	9.24	11.22

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
