# Peer review of "Conformation of Flexible and Semiflexible Chains Confined in Nanoposts Array of Various Geometries"

_polymers, 2020, doi:10.3390/polym12051064_

Round 1

Reviewer 1 Report

The authors address the study of the conformational properties of semiflexible and flexible polymers in an array of fixed posts bu using molecular dynamics simulations.

The study is interesting though some clarifications and improvements are required before the paper can be accepted for publication.

Here are some issues:

1. My first comment is about mathematical symbols. They appear not properly coded (e.g. Sp should be $S_p$ in latex to give the proper output in the text). The text is full of such typos.

2. Why do the authors apply the WCA potential also to bonded monomers since it is used to model excluded-volume effects?

3. line 128: R_=1.5. In which units? No length scale is yet defined.

4. lines 140-144: Formulas are not readable in the pdf file I received.

5. line 143: There is a missing symbol.

6. Which is the length of the simulated confined chains in terms of monomer number?

7. It is difficult to follow the overall discussion without having a visualizations of the system. I would suggest to add typical conformations of chains for some relevant cases.

8. In the spirit of helping the reader in understanding the study I would recommend to consider also the gyration radius of chains as well as the end-to-end orientation with respect to the longitudinal direction of posts.

After all the previous issued will be addressed, the paper might be reconsidered for publication in Polymers.

Author Response

  1. We apologize for the incorrectly coded mathematical symbols. However, the manuscript was written using Microsoft Office and it was also submitted in this form. The problem had to arise during the transcription to pdb format on the editorial platform after the submission. This problem will be discussed with the editorial office.
  2. In order to correctly represent the effective bond length between the effective monomers this type of potential requires to consider also the nonbonded interactions in addition to the FENE potential between connected effective monomers. This potential was designed by Binder et al and is frequently used in related studies, for instance, J. Chem. Phys. 143, 243102 (2015).
  3. We have added the length unit, now it writes .
  4. We again apologize for the problem associated with the file format conversion. The lines 140-144 including the formulas are as follows

“energy = with ui being the unit vector of i-th bond leads to the frequently adopted form of the bending potential energy

where q is the valence angle between two consecutive bonds in a chain (i.e., the complementary angle of ).”

  1. The same problem with the file format conversion. The symbol is q.
  2. The length of simulated confined chain is composed of 1000 effective monomers as we mention in line 151. We have extended the following sentence as follows

Since for the semiflexible chain one effective monomer represents 7.4 base pair this length approximately corresponds to 7.4 kbp.

  1. We have extended Figure 1 by a snapshot (b) “Cross-sectional view of a flexible chain confined in the nanopost array of Sp= 12 (b).
  2. We have not used the end-to-end distance to characterize the chain extension since the trend practically copied the trend of the axial chain extension. Thus, if using both quantities, one would be redundant. We have chosen the axial chain extension (span) since this is the quantity accessible in experimental measurements. The trend of the longitudinal component of the radius of gyration as well as the trend of the overall radius of gyration followed the trend of the axial chain extension. The trend of the lateral component of the radius of gyration followed the trend of the occupation number. We have extended the discussion at the end of 3.2. Section where we discuss the trends of the radius of gyration and its components. However, since these trends do not bring new information we prefer not to extend the text by respective plots.

Reviewer 2 Report

This manuscript continues a series of studies of the same authors on this very topic [49-51]

While they already investigated chain conformation for fixed post separation and fixed distance between post surfaces, 

here they study the case of constant post diameters. 

Although I find the overall topic interesting, the new work seems to provide only incremental new insights.

My main question is: In which sense is this novel and interesting? 

The authors claim in their conclusion on line 555 that the absolute value of the interstitial volumes and apertures play a role. 

Therefore, any variation leads to different results and potentially many manuscripts. But this can not be the aim! 

Take for example the axial extension shown in Fig 3. Fixing the post diameter (new case in black) is qualitatively quite similar to fixing the post separation (previous study in red). 

Other points: 

abstract: dc (line 18) not defined

Dp in Eq (2) vs dp in text and abstract

line 384: for the de Gennes D^-2/3 scaling one needs to identify regime of moderate confinement. 

But in Fig. 4, it seems to be the same as strong confinement in narrow channels assumed by Odijk.  

It should also be noted that the regime is way too small to infer a power-law. 

The structure factor in Eq (11), would it be helpful to consider the parallel and perpendicular components w.r.t post since structures are clearly anisotropic? 

The mathematical symbols - especially subscripts - and equations need better typesetting. 

Author Response

We have rephrased the conclusion and pointed out the main sense of this study as follows

This study completes the investigation of the effects arising from the different ways of geometry variation of the nanopost array on the structure of a confined flexible and semiflexible chain. The free energy arguments suggest that if the same confinement regime governs the conformation of a confined semiflexible chain the occupation number is supposed to be independent of the persistence length. In order to obtain more comprehensive insight into the effect arising from the chain stiffness, chains of different stiffness parameters need to be studied. Combination of narrow posts with large separations among them affects the conformation of the investigated confined semiflexible and flexible chain only negligibly.

Other points: 

abstract: dc (line 18) not defined

The dc parameter is defined in the following sentence (lines 14-16)

The free energy arguments based on an approximation of the array of nanopost to a composite of quasi-channels of diameter dc and quasi-slits of height wp provide semiqualitative explanations of observed structural behavior of both chains.

Dp in Eq (2) vs dp in text and abstract

Dp is the geometric diameter of the post in Equation 2 but the real diameter dp involves also the size of an effective monomer w ≅ 0.9σ representing the soft walls of nanopost. This is defined in the following sentence “The effective diameter of a post thus writes dp = Dp + w.“ in line 133.

line 384: for the de Gennes D^-2/3 scaling one needs to identify regime of moderate confinement.  But in Fig. 4, it seems to be the same as strong confinement in narrow channels assumed by Odijk.  It should also be noted that the regime is way too small to infer a power-law. 

In the sentence before Equation 10, we have defined the values of D for the extended as well as the classic de Gennes regime. The solid vertical line in Figure 4 demarcates the region with the dominance of single occupancy for both the flexible and semiflexible chain. It is meant as a line demarcating neither the Odijk regime nor the de Gennes regime. The meaning of this line is also described in the caption of Figure 4 as follow “The solid vertical line demarcates the region with the dominance of single-occupancy for both chains.

We have rephrased the text as follows: “In Figure 4, one can see that the axial chain extension at single occupancy satisfactorily follows the Odijk regime for the semiflexible chain and there is also an indication of the power low dependence of the axial chain extension for the flexible chain.

The structure factor in Eq (11), would it be helpful to consider the parallel and perpendicular components w.r.t post since structures are clearly anisotropic? 

We thanks the referee for this suggestion. Assuming the parallel and perpendicular components of the structure factor instead of the orientational average quantity would be very interesting. I would expect that the parallel component reflects the hierarchy of organization of the monomers within the individual interstitial volumes and contains the information on the size of interstitial volume. Thus, the similarity to the structure factor of a biaxially confined chain could be expected. The perpendicular component should reflect the periodicity introduced by the presence of the nanoposts and at larger values of the wavevectors, the organization of monomers within the blobs formed by the interstitial volumes. I haven’t encountered such a study in the related literature yet and this idea would deserve most likely a separate publication because the comprehensive study requires simulations of chains longer than 1000 monomers and larger scale of geometries of the nanoposts array with larger separations between posts.

The mathematical symbols - especially subscripts - and equations need better typesetting.

We apologize for the incorrectly coded mathematical symbols. However, the manuscript was written using Microsoft Office and it was also submitted in this form. The problem had to arise during the transcription to pdb format on the editorial platform after the submission. This problem will be discussed with the editorial office.

Round 2

Reviewer 1 Report

I am satisfied by the authors' reply. Panel (b) in Fig. 1 is missing. Please add it! The paper can then be accepted for publication.

Author Response

We thank the Reviewer for notifying us about the absence of panel b) in Figure 1. We have added the snapshot to Figure 1.  The snapshot is uploaded.

Reviewer 2 Report

The answers of the authors are satisfactory. 
